analytical chemistry/spectroscopy

[1]H-qNMR, fluticasone propionate, azelastine hydrochloride, nasal spray formulation

**Author for correspondence:**
Abdallah M. Zeid
e-mail: abdallah_zeid@hotmail.com

This article has been edited by the Royal Society of Chemistry, including the commissioning, peer review process and editorial aspects up to the point of acceptance.

# Quantitative proton nuclear magnetic resonance method for simultaneous analysis of fluticasone propionate and azelastine hydrochloride in nasal spray formulation

Amal A. El-Masry[1], Dalia R. El-Wasseef[1,3], Manal Eid[2], Ihsan A. Shehata[1] and Abdallah M. Zeid[2]

[1]Department of Medicinal Chemistry, and [2]Department of Pharmaceutical Analytical Chemistry, Faculty of Pharmacy, Mansoura University, 35516 Mansoura, Egypt
[3]Department of Pharmaceutical Chemistry, Faculty of Pharmacy, Delta University for Science and Technology, 35712 Gamasa, Egypt

(iD) AMZ, 0000-0001-5426-5993

A facile, rapid, accurate and selective quantitative proton nuclear magnetic resonance ([1]H-qNMR) method was developed for the simultaneous determination of fluticasone propionate (FLP) and azelastine hydrochloride (AZH) in pharmaceutical nasal spray for the first time. The [1]H-qNMR analysis of the studied analytes was performed using inositol as the internal standard and dimethyl sulfoxide-$d_6$ (DMSO-$d_6$) as the solvent. The quantitative selective proton signal of FLP was doublet of doublet at 6.290, 6.294, 6.316 and 6.319 ppm, while that of AZH was doublet at 8.292 and 8.310 ppm. The internal standard (inositol) produced a doublet signal at 3.70 and 3.71 ppm. The method was rectilinear over the concentration ranges of 0.25–20.0 and 0.2–15.0 mg ml$^{-1}$ for FLP and AZH, respectively. No labelling or pretreatment steps were required for NMR analysis of the studied analytes. The proposed [1]H-qNMR method was validated efficiently according to the International Council on Harmonisation guidelines in terms of linearity, limit of detection, limit of quantification, accuracy, precision, specificity and stability. Moreover, the method was applied to assay the analytes in their combined nasal spray formulation. The results ensured the linearity ($r^2 > 0.999$), precision (% RSD < 1.5), stability, specificity and selectivity of the developed method.

# 1. Introduction

Allergic rhinitis (hay fever) is an inflammatory complication caused by seasonal or perennial aeroallergens. Treatment of allergic rhinitis is mainly based on using intranasal glucocorticoids, oral and intranasal antihistamines, intranasal mast cell stabilizers, topical decongestant and leukotriene antagonist [1]. The reports revealed that the novel intranasal formulation containing fluticasone propionate (FLP) and azelastine hydrochloride (AZH) is the best option for treatment of allergic rhinitis because of its high efficiency in treatment of the symptoms compared to a monotherapy treatment with FLP or AZH [2].

Fluticasone propionate is chemically named as (S)-(fluoromethyl)-6$\alpha$,9-difluoro-11$\beta$, 17-dihydroxy-16$\alpha$-methyl-3-oxoandrosta-1,4-diene-17$\beta$-carbothioate, 17-propanoate (figure 1). It is a synthetic potent trifluorinated corticosteroid with a powerful anti-inflammatory activity that acts by preventing the release of inflammatory mediators in the body [3]. Azelastine hydrochloride is chemically named as (RS)-4-[(4-chlorophenyl) methyl]-2-(1-methylazepan-4-yl)-phthalazin-1-one (figure 1). It is a second-generation non-sedating antihistamine drug, which blocks $H_1$-receptor. It also possesses anti-inflammatory and mast cell stabilizing effects. It is used topically to relief allergic conditions such as rhinitis and conjunctivitis [4].

Reviewing the literature revealed that different analytical methods were established for determination of FLP and AZH either alone or in combination. These methods include: spectrophotometry [5–12], spectrofluorometry [13,14], electrochemical methods [15–17], TLC [18,19], HPLC [5,11,20–29] HPTLC [30–32], LC/MS/MS [33–39] and capillary electrophoresis [40,41]. It was found that only two spectrophotometric methods [42,43] and two HPLC methods [44,45] were reported for quantification of the binary mixture of FLP and AZH. However, no quantitative proton nuclear magnetic resonance ([1]H-qNMR) methods were reported for the simultaneous assay of FLP and AZH until now.

Therefore, the use of NMR to simultaneously determine FLP and AZH in their combined nasal spray preparation is highly advantageous. The advantages of the technique combine the great accuracy and the high precision which enable purity determination of pharmaceutical compounds efficiently [46]. Moreover, rapid analysis, direct measurement, facile sample preparation and non-destructive properties of the technique with the possibility to recover the analytes make it a suitable technique for the assay of pharmaceutical compounds. In addition, it does not need the pure target analyte as a reference sample for calibration because the signal intensity is directly proportional to the amount of proton atoms present, and no prior isolation is required in the analysis of multicomponent mixtures [47].

Herein, we have developed the first [1]H-qNMR method for simultaneous estimation of FLP and AZH in nasal spray formulation. The method showed superiority to the other reported ones in terms of simplicity, rapidity, reproducibility and direct analysis (no prior extraction or isolation). The analytical results of the proposed [1]H-qNMR method were statistically compared with the comparison HPLC method and the data confirmed no significant differences between the performance of the two methods. The proposed method was applied for analytical quality control of nasal spray preparation without any interference from formulation additives.

# 2. Experimental

## 2.1. Apparatus and condition

Quantitative determination of FLP and AZH binary mixture was performed by [1]H-qNMR. Spectra were recorded at 400 MHz using a Bruker Avance III spectrometer under the following acquisition and elaboration parameters: spectral width (15 ppm), frequency offset (6.175 ppm), acquisition time (4.08 s), flip angle (90°), pulse width (13.5 µs), sample temperature (20°C), dummy scans (2), relaxation delay (10 s), number of scans (64), sample spin (on), data points (65 536) and fixed receiver gain value (32 dB). On the NMR spectra, the chemical shifts were referenced to the doublet signal of inositol at 3.70, 3.71 ppm, doublet of doublet signal of FLP at 6.290, 6.294, 6.316, 6.319 ppm and doublet signal of AZH at 8.292, 8.310 ppm. The spectra were analysed using 0.3 Hz exponential line-broadening function, auto phase correction, integration and baseline correction.

The comparison HPLC method was carried out by a chromatographic system consisted of Knauer series P 6.1 L chromatograph equipped with a variable-wavelength UV–Vis detector (operated at 215 nm) and a Rheodyne injector valve bracket (fitted with a 20 µl sample loop). Collecting and processing data had been operated by Total Chrom Workstation (Massachusetts, USA). HPLC separation was performed on a shim-pack cyano column (250 × 4.6 mm) packed with 5 µm diameter particles, and 0.45 µm membrane filters (Millipore, Ireland) were used for filtration of the mobile phase.

fluticasone propionate                azelastine hydrochloride                inositol

**Figure 1.** Chemical structure of the studied analytes and the internal standard.

## 2.2. Materials and reagents

FLP and AZH certified as (99.50% and 99.80% purity, respectively) were kindly provided by European Egyptian Pharmaceuticals Industry, Alexandria, Egypt. The internal standard used for $^1$H-qNMR was inositol NF 12, which was kindly provided by October Pharma Industry S.A.E. 6th October City, Giza, Egypt (with 99.50% purity). The deuterated solvent used in the quantitative determination of this mixture was DMSO-$d_6$, which was purchased from Cambridge Isotope Laboratories, Inc. (D, 99.96%). Different deuterated solvents were investigated in the study such as chloroform-$d_1$ (D, 99.80%), acetone-$d_6$ (D, 99.90%) and deuterium oxide (D, 99.90%). All these solvents were purchased from Sigma-Aldrich. Phloroglucinol anhydrous (99.80% purity) was purchased from Chemi-pharm for Pharmaceutical Industries, Giza, Egypt. Formic acid (98.0% purity) was purchased from Sigma-Aldrich. Dymista nasal spray; labelled to contain 50 µg FLP and 137 µg AZH per spray, Batch no #546, manufactured by Cipla Ltd, Goa, India, M.L. for Meda Pharmaceuticals Inc. Somerset, NJ.

## 2.3. Preparation and analysis of standards

Standard stock solutions of (25.0, 20.0 and 100.0 mg ml$^{-1}$) for FLP, AZH and inositol, respectively, were prepared by dissolving 0.125 g FLP, 0.1 g AZH and 0.5 g inositol individually into three separate 5 ml volumetric flasks, and the volume was completed to the mark with DMSO-$d_6$. The volumetric flasks were sonicated for 20 min to ensure the homogeneity and dissolution. Increasing volumes of the standard solutions of FLP and AZH were quantitatively transferred into stoppered glass vials to provide solutions within the concentration range of 0.25–20.0 and 0.2–15.0 mg ml$^{-1}$ for FLP and AZH, respectively. Accurately measured volume of inositol internal standard in final concentration of 10.0 mg ml$^{-1}$ was then added to each vial. Thereafter, the volume was completed to 1.0 ml by DMSO-$d_6$. Quantitative NMR analysis was performed by taking 0.5 ml of each solution to 5 mm NMR tube, and proton spectra were carried out in triplicate for each concentration under the optimized acquisition and elaboration parameters. The absolute integral area ratio versus the final concentration of drugs were plotted to construct the calibration curves, and the regression equations were derived.

## 2.4. Analysis of FLP and AZH in their laboratory-prepared mixture

Standard stock solutions of both analytes were prepared by placing 0.0219 g FLP and 0.06 g AZH in stoppered glass vials and completed to 3 ml with DMSO-$d_6$ solvent to provide final concentrations of 7.3 and 20.0 mg ml$^{-1}$ for FLP and AZH, respectively. The ratio between FLP and AZH was kept 1 : 2.74 to mimic the medicinal ratio in nasal spray dosage. The same procedures mentioned above in the construction of calibration graphs were followed and the per cent of recoveries were measured.

## 2.5. Analysis of FLP and AZH in their nasal spray formulation

Twenty actuations of nasal spray suspension (Dymista) (1.0 mg FLP and 2.74 mg AZH) were actuated into a stoppered glass vial, where 50 µg of FLP and 137 µg of AZH were delivered after each actuation. Then, the content was left to dry overnight. One ml of DMSO-$d_6$ was then added and the flask was sonicated for 20 min. The previously performed steps in the construction of calibration graphs were followed. The nominal content of both drugs in its nasal spray dosage form was calculated.

## 2.6. Theory of quantitative NMR

There are two major types of relaxation processes: relaxation in the *z*-axis (spin-lattice relaxation) and relaxation in the *xy*-axis (spin-spin relaxation). Longitudinal (spin-lattice) relaxation time (*T*1) is the time required for the *z*-component of magnetization to reach approximately 63% of its maximum. This longitudinal relaxation time is influenced by different factors such as electron dipole interactions, dipole–dipole interactions and electric quadrupole interactions [48]. It can be calculated from the following equation:

$$\frac{M_z}{M_0} = (1 - e^{-TR/T1}), \quad \text{when } 90° \text{ pulse is applied,}$$

where $M_z$ is the longitudinal magnetization after pulse applying, $M_0$ is that in thermal equilibrium state and TR is the repetition time which is defined as the time between two excitation pulses (i.e. total time of pulse width, dead time, acquisition time and relaxation delay time). To recover 99.3% of $M_0$, TR must be five times the longest *T*1 of both the analyte and the internal standard.

The basic concept of qNMR is that the integrated signal area (*I*) in the NMR spectrum is directly proportional to the number of $^1$H nuclei on one molecule which is responsible for creating the resonance line: $I = Ks \times c \times n \times (1 - e^{-TR/T1})$.

When TR > *T*1 under quantitative conditions, $I = Ks \times c \times n$, where *K*s is the spectrometer constant, *c* is the molar concentration and *n* is the number of nuclei on one molecule. *K*s is dependent on many factors such as: pulse excitation, repetition time and broad band decoupling [48]. Quantitative determination of the analyte in qNMR relies on the use of the relative ratio of peak areas of analyte and standard material ($I_1/I_2$), where $I_1/I_2 = (c_1 \times n_1)/(c_2 \times n_2)$. *K*s is ignored in this ratio because it is constant in the same spectrum for all resonance lines.

Determination of the purity of the analyte can be also obtained from the NMR spectrum by using the subsequent equation [48]

$$\frac{I_1}{I_2} = \frac{c_1 \times n_1}{c_2 \times n_2} = \frac{((W_1 \times P_1 \times n_1)/(M_1 \times V_1))}{((W_2 \times P_2 \times n_2)/(M_2 \times V_2))}.$$

In the internal standard method, $V_1 = V_2$

$$\frac{I_1}{I_2} = \frac{((W_1 \times P_1 \times n_1)/M_1)}{((W_2 \times P_2 \times n_2)/M_2)}$$

and
$$P_1 = \left(\frac{I_1}{I_2}\right) \times \left(\frac{n_2}{n_1}\right) \times \left(\frac{M_1}{M_2}\right) \times \left(\frac{W_2}{W_1}\right)) \times P_2,$$

where $I_1$ and $I_2$ are the integral area, $n_1$ and $n_2$ are the number of the nuclei on one molecule, $M_1$ and $M_2$ are the molar mass, $W_1$ and $W_2$ are the gravimetric weight, $P_1$ and $P_2$ are the purity of both analyte and standard material, respectively.

# 3. Results and discussion

The $^1$H-qNMR method for determination of FLP and AZH was optimized carefully to assay the studied analytes efficiently in their combined dosage. The optimized parameters involved solvent selection, internal standard selection and investigation of other technical parameters of NMR such as number of scans, pulse angle and relaxation delay time. The optimum deuterated solvent selected for NMR analysis of the studied compounds was DMSO-$d_6$. Inositol NF 12 was chosen as the optimum internal standard which produced a doublet signal at 3.70 and 3.71 ppm. Such signal was efficiently resolved from the quantitative signals of the studied drugs. A doublet quantitative proton signal of AZH at 8.292, 8.310 ppm and a doublet of doublet signal of FLP at 6.290, 6.294, 6.316, 6.319 ppm were chosen for quantitative determination of both drugs as they were well separated and favourable for qNMR application under the optimum acquisition parameters as shown in figure 2.

## 3.1. Deuterated solvent selection

Different deuterated solvents were screened in this study to select the optimum one that allows quantitative determination of both drugs without interference. The studied solvents included DMSO-$d_6$, chloroform-$d_1$, acetone-$d_6$ and deuterium oxide. Both chloroform-$d_1$ and acetone-$d_6$ are highly volatile, so when drugs were

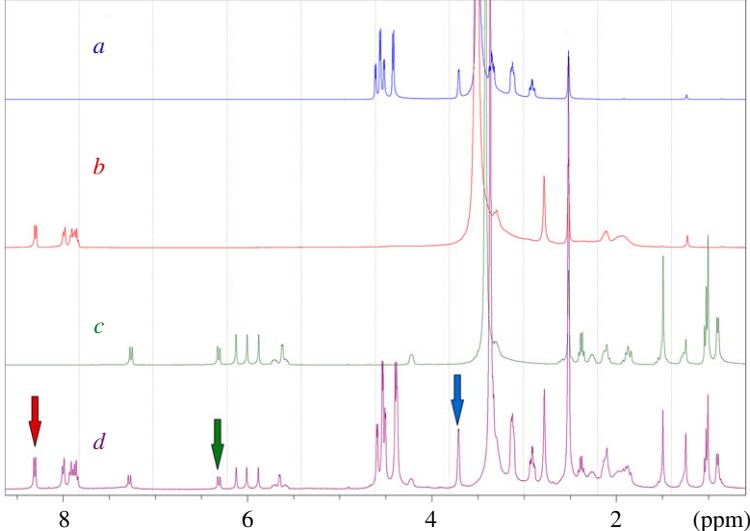

**Figure 2.** ¹H-NMR spectra of (*a*) inositol NF 12, (*b*) azelastine HCl (AZH), (*c*) fluticasone propionate (FLP) and (*d*) a mixture of AZH and FLP (containing inositol NF 12 as internal standard), in dimethylsulfoxide-*d₆* (DMSO- *d₆*) under the optimum acquisition and elaboration conditions.

dissolved in such solvents, the final volumes of the solutions were varied, and the concentrations of drugs were inaccurate. Deuterium oxide was also excluded because of poor solubility of the studied drugs in deuterium oxide. The results indicated that DMSO-*d₆* is the most appropriate deuterated solvent as it is non-volatile at room temperature, it allowed good solubility of the studied drugs, and its signal at 2.5 ppm did not overlap with the quantitative NMR signals of FLP, AZH and inositol.

## 3.2. Internal standard selection

Different compounds were tested to select the most appropriate internal standard. The investigated compounds included phloroglucinol, formic acid and inositol NF 12. Inositol NF 12 was selected as the most suitable one because its doublet signal at 3.70 and 3.71 ppm did not interfere with the integration region of quantitative protons signals of FLP (6.290, 6.294, 6.316, 6.319 ppm) and AZH (8.292, 8.310 ppm).

## 3.3. Optimization of technical NMR parameters

Different parameters that contribute to the efficiency of ¹H-qNMR technique were studied to select the optimum conditions that provide efficient quantification with satisfactory results. These parameters included number of scans, pulse angle and relaxation delay time.

### 3.3.1. Number of scans

Number of scans is one of the main parameters that can be adjusted to improve signal to noise. Different number of scans (16, 32, 64 and 128) were investigated in this study. It was found that increasing the number of scans resulted in increased scanning time and improved sensitivity. Each experiment is repeated three times and the average was abridged as shown in figure 3*a*. Although the scan number of 128 resulted in higher integral area of both compounds (figure 3*a*), the results were not reproducible. Therefore, a scan number of 64 was selected as the optimum scan number that provided high sensitivity and adequate reproducibility.

### 3.3.2. Pulse angle

Different values of pulse angles (10°, 30°, 60° and 90°) were investigated, keeping number of scans at 64 and relaxation delay time at 10 s. The resulting absolute integral area over these angles was plotted in (figure 3*b*). The results indicated that a pulse angle of 90° is the most appropriate one as it showed best absolute integral area for both drugs.

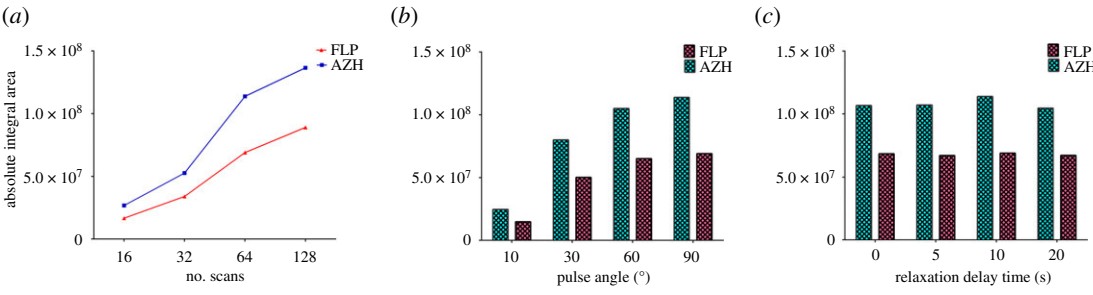

**Figure 3.** Influence of (*a*) number of scans, (*b*) pulse angle and (*c*) relaxation delay time, on the absolute integral area of the selected signals of FLP and AZH in ¹H-NMR spectra.

**Table 1.** Performance data for the determination of the FLP and AZH through the laboratory-performed ¹H-qNMR method.

| parameter | FLP | AZH |
|---|---|---|
| concentration range (mg ml⁻¹) | 0.25–20.0 | 0.20–15.0 |
| correlation coefficient | 0.9999 | 0.9999 |
| slope | 0.02 | 0.03 |
| intercept | $64 \times 10^{-4}$ | $1 \times 10^{-4}$ |
| LOD (mg ml⁻¹) | 0.04 | 0.02 |
| LOQ (mg ml⁻¹) | 0.12 | 0.05 |
| $S_{y/x}$ | $43 \times 10^{-5}$ | $27 \times 10^{-5}$ |
| $S_a$ | $21 \times 10^{-5}$ | $13 \times 10^{-5}$ |
| $S_b$ | $2 \times 10^{-5}$ | $2 \times 10^{-5}$ |
| % RSD | 0.91 | 1.04 |
| % error = $(s.d./\sqrt{n})$ | 0.32 | 0.37 |

### 3.3.3. Relaxation delay time

Different relaxation delay times (0, 5, 10 and 20 s) were investigated, keeping number of scans at 64 and pulse angle at 90°. The absolute integral area was recorded in each adjusted relaxation delay time as shown in figure 3*c*. The optimum relaxation delay time was found to be 10 s because it was enough to ensure perfectly longitudinal relaxation between two adjacent pulses and gave best signal resolution and well quantitative assay.

## 3.4. Method validation

The proposed method was validated according to ICH Q2 (R1) guidelines [49] for the following parameters:

### 3.4.1. Linearity and range

Linearity of the proposed method was illustrated depending on the fact discussed above where the integrated signal area in the NMR spectrum is directly proportional to the nuclei number. Different standard solutions for both drugs FLP and AZH at eight different concentrations ranging from 0.25 to 20.0 and 0.2 to 15.0 µg ml⁻¹ for FLP and AZH, respectively, with internal standard were prepared, a perfect linear relationship between an absolute integral area ratio (ratio between area of drug and internal standard (inositol)) and drug concentration was achieved. The resulting data were statistically analysed [50] and showed low scattering of points around the calibration curve, low percentage relative standard deviation (% RSD did not exceed 2.0), low percentage relative error (% error) with high value of correlation coefficient (*r*). The analytical performance data are presented in table 1. Linear regression

**Table 2.** Comparative analytical data for determination of FLP and AZH in pure form by the proposed [1]H-qNMR method and comparison HPLC method.

| drug | proposed [1]H-qNMR method | | | comparison HPLC method [45] |
| | amount taken (mg ml$^{-1}$) | amount found (mg ml$^{-1}$) | % recovery[a] | % recovery[a] |
|---|---|---|---|---|
| FLP | 0.25 | 0.25 | 98.80 | 100.19 |
| | 0.30 | 0.30 | 100.33 | 99.26 |
| | 0.80 | 0.82 | 102.00 | 101.71 |
| | 1.00 | 1.00 | 100.10 | 99.65 |
| | 5.00 | 4.98 | 99.52 | |
| | 10.00 | 10.04 | 100.36 | |
| | 15.00 | 14.98 | 99.84 | |
| | 20.00 | 20.04 | 100.18 | |
| $\bar{X} \pm$ s.d. | | 100.14 $\pm$ 0.91 | | 100.20 $\pm$ 1.08 |
| t-test | | 0.1 (2.23)[b] | | |
| F-value | | 1.39 (8.89)[b] | | |
| AZH | 0.20 | 0.20 | 98.50 | 98.84 |
| | 0.30 | 0.29 | 98.00 | 99.13 |
| | 0.80 | 0.81 | 101.38 | 101.04 |
| | 1.00 | 1.00 | 99.60 | 99.43 |
| | 5.00 | 4.99 | 99.78 | |
| | 8.00 | 8.01 | 100.18 | |
| | 10.00 | 10.00 | 99.96 | |
| | 15.00 | 14.99 | 99.95 | |
| $\bar{X} \pm$ s.d. | | 99.67 $\pm$ 1.04 | | 99.61 $\pm$ 0.98 |
| t-test | | 0.09 (2.23)[b] | | |
| F-value | | 1.11 (8.89)[b] | | |

[a]Each result is the mean recovery of three separate determinations.
[b]Figures between brackets are the tabulated t- and F-values at $p = 0.05$.

analysis of the data obtained using the developed method gave the following equations:

$$I = 64 \times 10^{-4} + 0.02C \ (r = 0.9999) \quad \text{for FLP}$$

and

$$I = 1 \times 10^{-4} + 0.03C \ (r = 0.9999) \quad \text{for AZH,}$$

where $I$ is absolute integral area ratio, $C$ is the concentration of the drug (mg ml$^{-1}$) and $r$ is correlation coefficient.

### 3.4.2. Limits of detection and quantification

In the laboratory-performed method, limits of detection (LOD) and limits of quantification (LOQ) were calculated according to ICH Q2 (R1) recommendations [49] using the following equations:

$$\text{LOD} = \frac{3.3 S_a}{b}$$

and

$$\text{LOQ} = \frac{10 S_a}{b},$$

where $S_a$ is the standard deviation of the intercept and $b$ is the slope of the calibration curve.

**Table 3.** Intra-day and inter-day precision data for the assay of FLP and AZH by the proposed ¹H-qNMR method.

| parameters | | FLP concentration (mg ml⁻¹) | | | AZH concentration (mg ml⁻¹) | | |
| --- | --- | --- | --- | --- | --- | --- | --- |
| | | 1.0 | 5.0 | 10.0 | 1.0 | 5.0 | 10.0 |
| intra-day | % found[a] | 101.27 | 100.12 | 101.57 | 99.64 | 100.29 | 98.53 |
| | | 100.06 | 99.52 | 100.36 | 100.36 | 99.78 | 99.96 |
| | | 101.22 | 100.00 | 100.96 | 100.72 | 99.28 | 99.25 |
| | X̄ | 100.85 | 99.88 | 100.96 | 100.24 | 99.78 | 99.25 |
| | ±s.d. | 0.68 | 0.32 | 0.60 | 0.55 | 0.50 | 0.72 |
| | % RSD | 0.68 | 0.32 | 0.60 | 0.55 | 0.50 | 0.72 |
| | % error | 0.39 | 0.18 | 0.34 | 0.32 | 0.29 | 0.42 |
| inter-day | % found[a] | 101.08 | 100.48 | 100.96 | 98.57 | 100.29 | 98.53 |
| | | 100.06 | 99.52 | 100.36 | 100.36 | 99.78 | 99.96 |
| | | 99.46 | 99.76 | 99.16 | 99.64 | 98.04 | 100.39 |
| | X̄ | 100.20 | 99.92 | 100.16 | 99.52 | 99.37 | 99.63 |
| | ±s.d. | 0.82 | 0.50 | 0.92 | 0.90 | 1.20 | 0.98 |
| | % RSD | 0.82 | 0.50 | 0.92 | 0.90 | 1.20 | 0.98 |
| | % error | 0.47 | 0.29 | 0.53 | 0.52 | 0.69 | 0.57 |

[a]Each result is the mean recovery of three individual determinations.

**Table 4.** Sample stability data for the assay of FLP and AZH by the proposed ¹H-qNMR method.

| time (hour) | per cent assay of the sample solution | |
| --- | --- | --- |
| | FLP (%) | AZH (%) |
| 0 | 100.36 | 99.96 |
| 6 | 100.42 | 99.86 |
| 12 | 100.30 | 99.57 |
| 24 | 100.06 | 99.93 |
| mean (X̄) | 100.29 | 99.83 |
| RSD % | 0.16 | 0.18 |

The results indicated an adequate sensitivity of the proposed method (LOD ≤ 0.04 and LOQ ≤ 0.12 mg ml⁻¹) that is suitable for the simultaneous estimation of FLP and AZH in their combined pharmaceutical dosage (table 1).

### 3.4.3. Accuracy and precision

Accuracy is the closeness of agreement between the measured value from the experimental proposed method and the known reference value. An HPLC comparison method [45] was applied to compare the analytical results of both drugs in pure form with those obtained from the ¹H-qNMR proposed method. Separation of FLP and AZH in the HPLC reference method was performed by using (250 × 4.6 mm, 5 μm particle size) cyano column with mobile phase consisted of 55 : 45 (v/v) mixture of phosphate buffer and acetonitrile; elution of both drugs was under flow rate of 1 ml min⁻¹ with UV detection at 215 nm. Three different concentrations of both FLP and AZH were measured in triplicate. Standing on peak area and drug concentration, the per cent of recoveries for each concentration for each drug were measured and the average recoveries (% purity) are abridged in table 2. Satisfactory results were obtained, as there was no significant difference by using both Student's *t*-test and variance ratio *F*-test, as shown in table 2.

Intra-day and inter-day precision of ¹H-qNMR proposed method were estimated by triplicate assay of FLP and AZH (each in pure form), using three different concentrations (1.0, 5.0 and 10.0 mg ml⁻¹) for

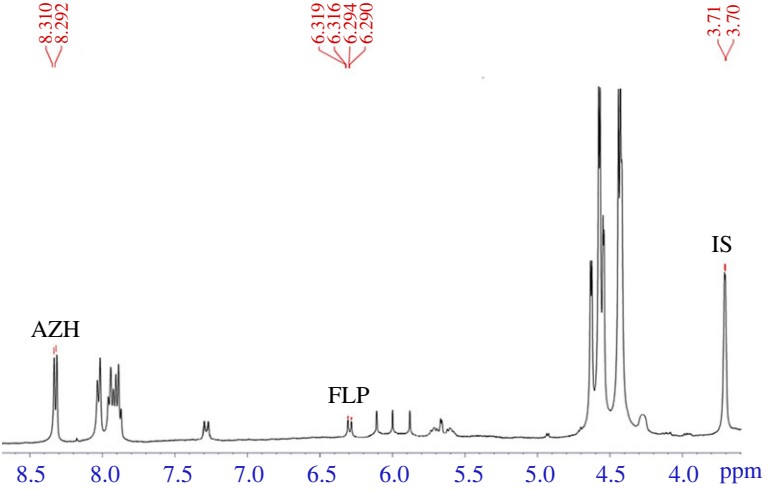

**Figure 4.** ¹H-NMR spectrum of fluticasone propionate (FLP) and azelastine hydrochloride (AZH) in their laboratory synthetic mixture using inositol NF 12 as the internal standard (IS) and dimethylsulfoxide-$d_6$ (DMSO-$d_6$) as the solvent.

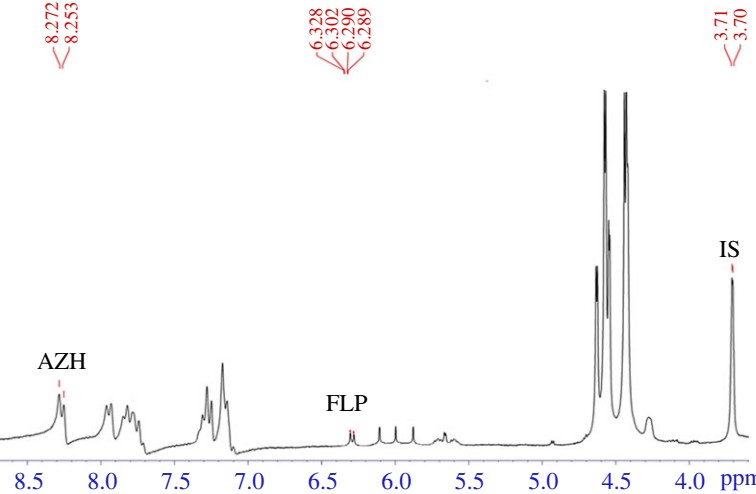

**Figure 5.** ¹H-NMR spectrum of fluticasone propionate (FLP) and azelastine hydrochloride (AZH) in Dymista® nasal spray using inositol NF 12 as the internal standard (IS) and dimethyl sulfoxide-$d_6$ (DMSO-$d_6$) as the solvent.

both drugs in 1 day and for 3 successive days. Low values of relative standard deviations confirmed the good reproducibility and precision of the performed method, as summarized in table 3.

### 3.4.4. Specificity and selectivity

Specificity of the proposed method was estimated to ensure the absence of any interference caused by the excipients or the solvent. Proton NMR spectra of DMSO-$d_6$ (blank), inositol (internal standard), drug standards and sample solutions were constructed individually. It was found that there were well-separated signals of internal standard and both drugs (FLP and AZH) without any overlap at the integral signals of FLP (6.290, 6.294, 6.316 and 6.319 ppm), AZH (8.292 and 8.310 ppm) and inositol (3.70 and 3.71 ppm), as shown in figure 2, which proved the good specificity and selectivity of the method.

### 3.4.5. Drug stability

Studying the stability of drugs is an important factor to estimate the time span between collection of the sample and its analysis. This test was applied by analysis of the same sample solution at four different time intervals (0, 6, 12 and 24 h) at ambient temperature. The result ensured the stability of both drugs for this period, where the calculated relative standard deviation percentages were 0.16 and 0.18 for FLP and AZH, respectively, as shown in table 4.

**Table 5.** Comparative resultant data from simultaneous determination of FLP and AZH in their laboratory-prepared mixture by the proposed [1]H-qNMR and comparison HPLC method.

| drug | proposed [1]H-qNMR method | | | comparison HPLC method [45] |
| --- | --- | --- | --- | --- |
| | amount taken (mg ml[−1]) | amount found (mg ml[−1]) | % recovery[a] | % recovery[a] |
| FLP | 0.37 | 0.37 | 100.84 | 99.99 |
| | 1.83 | 1.80 | 98.70 | 102.37 |
| | 2.92 | 2.90 | 99.23 | 101.62 |
| X̄ ± s.d. | | 99.59 ± 1.12 | | 101.33 ± 1.22 |
| t-test | | 1.82 (2.78)[b] | | |
| F-value | | 1.19 (19)[b] | | |
| AZH | 1.00 | 1.00 | 100.36 | 99.89 |
| | 5.00 | 4.99 | 99.78 | 100.33 |
| | 8.00 | 7.95 | 99.37 | 101.43 |
| X̄ ± s.d. | | 99.84 ± 0.50 | | 100.55 ± 0.79 |
| t-test | | 1.32 (2.78)[b] | | |
| F-value | | 2.54 (19)[b] | | |

[a]Each result is the mean recovery of three separate determinations.
[b]Figures between brackets are the tabulated t- and F-values at $p = 0.05$.

**Table 6.** Comparative resulting data from simultaneous determination of FLP and AZH in their pharmaceutical nasal spray suspension by the proposed [1]H-qNMR and comparison HPLC method.

| drug | proposed [1]H-qNMR method | | | | | | comparison HPLC method [45] | |
| --- | --- | --- | --- | --- | --- | --- | --- | --- |
| | amount taken (mg ml[−1]) | | amount found (mg ml[−1]) | | % recovery[b] | | % recovery[b] | |
| Dymista® nasal spray[a] FLP/AZH | FLP | AZH | FLP | AZH | FLP | AZH | FLP | AZH |
| | 0.40 | 1.096 | 0.39 | 1.11 | 98.64 | 100.98 | 99.16 | 99.96 |
| | 0.60 | 1.644 | 0.60 | 1.63 | 100.63 | 98.98 | 98.82 | 100.13 |
| | 0.80 | 2.192 | 0.79 | 2.18 | 98.34 | 99.42 | 100.89 | 101.75 |
| | FLP | | AZH | | | | | |
| X̄ ± s.d. | 99.21 ± 1.24 | | 99.79 ± 1.05 | | | | 99.62 ± 1.11 | 100.61 ± 0.99 |
| t-test | 0.44 (2.78)[c] | | 0.99 (2.78)[c] | | | | | |
| F-value | 1.26 (19)[c] | | 1.13 (19)[c] | | | | | |

[a]Dymista nasal spray suspension; labelled to contain 50 µg FLP and 137 µg AZH per each spray, Batch no # 546, manufactured by Cipla Ltd, India, M.L. for Meda Pharmaceuticals Inc. Somerset, NJ.
[b]Each result is the mean recovery of three separate determinations.
[c]Figures between brackets are the tabulated t- and F-values at $p = 0.05$.

## 3.5. Application

### 3.5.1. Assay of FLP and AZH in their laboratory-prepared mixture and in their nasal spray dosage

The proposed NMR method was applied to assay the studied analytes in their laboratory-prepared mixtures and in their nasal spray dosage. Synthetic mixtures of FLP and AZH with three different concentrations were prepared with constant ratio of (1 : 2.74) (FLP : AZH) to mimic the ratio in their pharmaceutical preparation. Analytical application of the proposed [1]H-qNMR method on these

synthetic mixtures and on their pharmaceutical nasal spray was performed efficiently with high specificity and selectivity (figures 4 and 5). No significant overlap from formulations' excipients were observed and the resulting data were statistically analysed using student's $t$-test and variance ratio $F$-test [50]. These results were compared with the comparison HPLC method [45] where small percentage value of relative standard deviation and high recoveries percentage value ensured the acceptable analytical application of the proposed method for quality control of FLP and AZH in their pharmaceutical nasal spray as shown in tables 5 and 6.

## 4. Conclusion

A simple, rapid, accurate and reliable $^1$H-qNMR method was established for the simultaneous determination of FLP and AZH binary mixture, which is used for treatment of allergic rhinitis and relieving its symptoms. The method is the first $^1$H-qNMR with non-destructive procedure that analyse both FLP and AZH with no prior extraction or pretreatment. The proposed $^1$H-qNMR method was validated according to ICH guidelines and efficiently applied to assay the studied compounds in pure form, synthetic mixture and nasal spray dosage form, using inositol as the internal standard and DMSO-$d_6$ as the deuterated solvent. The superiority of the proposed qNMR method compared to other reported methods was based on rapid analysis, high flexibility in choosing reference substance (no need to use pure analyte), ability of recovering the analyte (non-destructive method), and ability to assay multicomponent mixtures without tedious separation processes. All these advantages enabled $^1$H-qNMR to be applied in quality control analysis of the studied compounds in their dosage form. In addition, it will open a wide area for many future applications such as pharmacokinetics studies, forensic analysis and environmental analysis.

Data accessibility. Data are available from the Dryad Digital Repository https://doi.org/10.5061/dryad.rn8pk0p98 [51].
Authors' contributions. A.A.E. carried out the laboratory work, participated in data analysis, participated in the design of the study and drafted the manuscript; D.R.E. carried out the statistical analyses, conceived of the study, designed the study and helped draft the manuscript. M.E. participated in the design of the study, participated in the statistical analyses and revised the manuscript; I.A.S. designed the study, coordinated the study and revised the manuscript; A.M.Z. participated in data analysis, participated in the design of the study, validated the study and critically revised the manuscript. All authors gave final approval for publication.
Competing interests. The authors declare that they have no conflict of interest.
Funding. No funding supported this research.
Acknowledgements. The authors are thankful to Mohamed A. Sabry who is one of the members in NMR Unit, Faculty of Pharmacy, Mansoura University.

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
