## [Peer Review File · Royal Society Open Science]

Review History

RSOS-210483.R0 (Original submission)

Review form: Reviewer 1

Is the manuscript scientifically sound in its present form?

Yes

Are the interpretations and conclusions justified by the results?

Yes

Is the language acceptable?

Yes

Do you have any ethical concerns with this paper?

No

Have you any concerns about statistical analyses in this paper?

No

Recommendation?

Accept with minor revision (please list in comments)

Comments to the Author(s)

Reviewer comments to author:

The authors described a new quantitative proton nuclear magnetic resonance (1H-qNMR) method for the simultaneous determination of fluticasone propionate and azelastine hydrochloride in pharmaceutical nasal spray. The 1H-qNMR analysis of the studied analytes was performed using inositol as the internal standard and dimethyl sulfoxide-d₆ as the solvent. In general, the manuscript is written well with a good discussion of the analytical method. The method was validated and applied for the first time to assay fluticasone propionate and azelastine hydrochloride in nasal spray with high % recovery and low values of %RSD. In addition, the method did not involve tedious labeling or pretreatment steps. Therefore, I recommend acceptance of the manuscript after responding to the comments below:

- 1- Introduction section, page 3, lines 21 and 26, replace (Fig. 1a) and (Fig. 1b) with (Fig. 1), as there is no (a or b) in Fig. 1.
- 2- Introduction section, page 4, line 12, the authors used one combined dosage form (Dymista® nasal spray), so it would be better to write "formulation" instead of "formulations".
- 3- Fig. 2 caption, page 9, line 54, replace the sentence "containing inositol NF internal standard" to be "containing inositol NF 12 as internal standard".
- 4- Results and discussion: page 9, line 10, make "d₆" Italic in "acetone-d₆".
- 5- Fig. 3: It seems that it is not a correct figure format. So, revise the figure format and adjust the "x" symbol.
- 6- Page 12, line 53 (accuracy and precision subsection): Using three different concentrations (0.89, 4.44, 8.88 mg/mL) for both drugs in one day and for three successive days. These three concentrations are not the same concentrations mentioned in Table 3 (1.0, 5.0, 10.0 mg/mL). Please revise and correct.
- 7- What is the rationale for selecting such three concentrations for intra-day and inter-day precision?
- 8- Conclusion: page 20, line 16, add full stop after the word "pretreatment", and at line 34, replace "dosage forms" with "dosage form".
- 9- The references should be revised again carefully and formatted properly according to the guidelines of Royal Society Open Science (it is preferable to use a software like EndNote or other available software).

Review form: Reviewer 2

Is the manuscript scientifically sound in its present form?

Yes

Are the interpretations and conclusions justified by the results?

Yes

Is the language acceptable?

Yes

Do you have any ethical concerns with this paper?

No

Have you any concerns about statistical analyses in this paper?

No

Recommendation?

Accept as is

Comments to the Author(s)

Current form of manuscript is acceptable.

Decision letter (RSOS-210483.R0)

Dear Dr Zeid:

Title: Quantitative proton NMR method for simultaneous determination of fluticasone propionate and azelastine hydrochloride in nasal spray formulation
Manuscript ID: RSOS-210483

Thank you for submitting the above manuscript to Royal Society Open Science. On behalf of the Editors and the Royal Society of Chemistry, I am pleased to inform you that your manuscript will be accepted for publication in Royal Society Open Science subject to minor revision in accordance with the referee suggestions. Please find the reviewers' comments at the end of this email.

The reviewers and handling editors have recommended publication, but also suggest some minor revisions to your manuscript. Therefore, I invite you to respond to the comments and revise your manuscript.

Because the schedule for publication is very tight, it is a condition of publication that you submit the revised version of your manuscript before 04-Jun-2021. Please note that the revision deadline will expire at 00.00am on this date. If you do not think you will be able to meet this date please let me know immediately.

1) A text file of the manuscript (tex, txt, rtf, docx or doc), references, tables (including captions) and figure captions. Do not upload a PDF as your "Main Document".

- 2) A separate electronic file of each figure (EPS or print-quality PDF preferred (either format should be produced directly from original creation package), or original software format)
- 3) Included a 100 word media summary of your paper when requested at submission. Please ensure you have entered correct contact details (email, institution and telephone) in your user account
- 4) Included the raw data to support the claims made in your paper. You can either include your data as electronic supplementary material or upload to a repository and include the relevant doi within your manuscript
- 5) All supplementary materials accompanying an accepted article will be treated as in their final form. Note that the Royal Society will neither edit nor typeset supplementary material and it will be hosted as provided. Please ensure that the supplementary material includes the paper details where possible (authors, article title, journal name).

Kind regards,
 Dr Laura Smith
 Publishing Editor, Journals

RSC Associate Editor:
 Comments to the Author:
 (There are no comments.)

RSC Subject Editor:
 Comments to the Author:
 (There are no comments.)

Reviewer comments to Author:

Reviewer: 1

Comments to the Author(s)

Reviewer comments to author:

The authors described a new quantitative proton nuclear magnetic resonance (¹H-qNMR) method for the simultaneous determination of fluticasone propionate and azelastine hydrochloride in pharmaceutical nasal spray. The ¹H-qNMR analysis of the studied analytes was performed using inositol as the internal standard and dimethyl sulfoxide-d₆ as the solvent. In general, the manuscript is written well with a good discussion of the analytical method. The method was validated and applied for the first time to assay fluticasone propionate and azelastine hydrochloride in nasal spray with high % recovery and low values of %RSD. In addition, the method did not involve tedious labeling or pretreatment steps. Therefore, I recommend acceptance of the manuscript after responding to the comments below:

1- Introduction section, page 3, lines 21 and 26, replace (Fig. 1a) and (Fig. 1b) with (Fig. 1), as there is no (a or b) in Fig. 1.

2- Introduction section, page 4, line 12, the authors used one combined dosage form (Dymista® nasal spray), so it would be better to write "formulation" instead of "formulations".

3- Fig. 2 caption, page 9, line 54, replace the sentence "containing inositol NF internal standard" to be "containing inositol NF 12 as internal standard".

4- Results and discussion: page 9, line 10, make "d₆" Italic in "acetone-d₆".

5- Fig. 3: It seems that it is not a correct figure format. So, revise the figure format and adjust the "x" symbol.

6- Page 12, line 53 (accuracy and precision subsection): Using three different concentrations (0.89, 4.44, 8.88 mg/mL) for both drugs in one day and for three successive days. These three concentrations are not the same concentrations mentioned in Table 3 (1.0, 5.0, 10.0 mg/mL).

Please revise and correct.

7- What is the rationale for selecting such three concentrations for intra-day and inter-day precision?

8- Conclusion: page 20, line 16, add full stop after the word "pretreatment", and at line 34, replace "dosage forms" with "dosage form".

9- The references should be revised again carefully and formatted properly according to the guidelines of Royal Society Open Science (it is preferable to use a software like EndNote or other available software).

Reviewer: 2

Comments to the Author(s)

Current form of manuscript is acceptable.

Author's Response to Decision Letter for (RSOS-210483.R0)

See Appendix A.

Decision letter (RSOS-210483.R1)

Dear Dr Zeid:

Title: Quantitative proton NMR method for simultaneous determination of fluticasone propionate and azelastine hydrochloride in nasal spray formulation
Manuscript ID: RSOS-210483.R1

It is a pleasure to accept your manuscript in its current form for publication in Royal Society Open Science. The chemistry content of Royal Society Open Science is published in collaboration with the Royal Society of Chemistry.

RSC Associate Editor
Comments to the Author:
(There are no comments.)

Reviewer(s)' Comments to Author:

Appendix A

Dear Editor,

First, I would like to thank you and the reviewers for the valuable and helpful comments. Here is a point-by-point response to the reviewer comments. All changes in the manuscript were made by **Track Changes** in the revised manuscript.

Reviewer 1:

1- Introduction section, page 3, lines 21 and 26, replace (Fig. 1a) and (Fig. 1b) with (Fig. 1), as there is no (a or b) in Fig. 1.

- **It was corrected by omitting the letters ‘a’ and ‘b’ in Fig. 1a and Fig. 1b as recommended.**

2- Introduction section, page 4, line 12, the authors used one combined dosage form (Dymista® nasal spray), so it would be better to write “formulation” instead of “formulations”.

- **The letter “s” was omitted from the word “formulations” as recommended.**

3- Fig. 2 caption, page 9, line 54, replace the sentence “containing inositol NF internal standard” to be “containing inositol NF 12 as internal standard”.

- **The sentence “containing inositol NF internal standard” in Fig. 2 caption was replaced with “containing inositol NF 12 as internal standard” as recommended.**

4- Results and discussion: page 9, line 10, make “d6” Italic in “acetone-d6”.

- **“d6” in “acetone-d6” was written in Italic as recommended.**

5- Fig. 3: It seems that it is not a correct figure format. So, revise the figure format and adjust the “×” symbol.

- **The format of Fig. 3 was corrected and the “x” symbol in the figure was adjusted as recommended.**

6- Page 12, line 53 (accuracy and precision subsection): Using three different concentrations (0.89, 4.44, 8.88 mg/mL) for both drugs in one day and for three successive days. These three concentrations are not the same concentrations mentioned in Table 3 (1.0, 5.0, 10.0 mg/mL). Please revise and correct.

- **It was a typing mistake. The three used concentrations were (1.0, 5.0, 10.0 mg/mL), so it was corrected in the text.**

7- What is the rationale for selecting such three concentrations for intra-day and inter-day precision?

- **The ICH recommended 9 determinations covering the specified range (e.g. 3 concentrations / 3 replicates each) but it did not determine these values except to be within the range. So, these concentrations were taken within the specified range.**

8- Conclusion: page 20, line 16, add full stop after the word “pretreatment”, and at line 34, replace “dosage forms” with “dosage form”.

- A full stop (period) was added at the end of the sentence after the word “pretreatment”, and “Dosage forms” was replaced with “dosage form” as recommended.

9- The references should be revised again carefully and formatted properly according to the guidelines of Royal Society Open Science (it is preferable to use a software like EndNote or other available software).

- The references were revised again and formatted according to the guidelines of Royal Society Open Science Journal using EndNote software.

Reviewer 2:

- Thank you for your recommendation.

Moreover, the tables and figure captions were moved after the references, and the figures were submitted in separate files according to the journal guidelines.

Finally, I would like to thank you again for these valuable comments that help in improving the quality of the manuscript. The changes were made by **Track Changes** in the revised manuscript.

I look forward to hearing from you.

Best regards.

Sincerely Yours,

Abdallah M. Zeid, Ph.D.

Assistant Professor, Dept. of Pharm. Anal. Chem.,
Faculty of Pharmacy, Mansoura University, Egypt.

Former researcher

Department of Biomolecular Engineering,
Graduate School of Engineering, Nagoya University, Japan.

Phone: +20-102-779-2538, Fax: +20-50-224-7496;

E-mail: Abdallah_zeid@hotmail.com